Evaluating cognitive and affective abilities among medical students: behavioural and medicolegal perspectives

Amin Shaimaa Nasr 1 2 shaimaa@hu.edu.jo
Al-Jussani Ghada Nazar 3
http://orcid.org/0000-0003-3634-8909 S Hassan Sherif 4
http://orcid.org/0000-0002-6104-562X Sharif Asmaa F. 5
Ismail Ahmed A. 6 7
Badie Taher Dina 8
http://orcid.org/0000-0002-1231-1071 Aolymat Iman 1
El-Akabawy Gehan 9 10 11
Zayed Abeer Ahmed 12
1 Department of Anatomy, Physiology and Biochemistry, Faculty of Medicine, The Hashemite University , Zarqa , Jordan
2 Physiology Department, Faculty of Medicine, Cairo University , Cairo , Egypt
3 Department of Microbiology, Pathology and Forensic Medicine, Faculty of Medicine, The Hashemite University , Zarqa , Jordan
4 Department of Internal Medicine, University of California , Riverside, California , United States
5 Department of Forensic Medicine and Clinical Toxicology, Faculty of Medicine, Tanta University , Tanta , Egypt
6 Community Medicine and Public Health Department, Faculty of Medicine, Menoufia University , Shebin Elkom , Egypt
7 Kansas Department of Health and Environment, Topeka , Kansas , USA
8 Department of Psychiatry, Kasr al Ainy, Faculty of Medicine, Cairo University , Cairo , Egypt
9 Department of Basic Medical Sciences, College of Medicine, Ajman University , Ajman , United Arab Emirates
10 Centre of Medical and Bio-allied Health Sciences Research, Ajman University , Ajman , United Arab Emirates
11 Department of Anatomy and Embryology, Faculty of Medicine, Menoufia University , Menoufia , Egypt
12 Department of Forensic Medicine & Clinical Toxicology, Faculty of Medicine, Cairo University , Cairo , Egypt
Khosravi Mohsen
Electronic publication date: 2024 Feb 2
Publication date: 2024
Volume: 12
Electronic Location ID: e16864
Received 2023 Nov 6; Accepted 2024 Jan 9
Copyright: © 2024 Amin et al.
Copyright year: 2024
Copyright holder: Amin et al.
License: This is an open access article distributed under the terms of the Creative Commons Attribution License, which permits unrestricted use, distribution, reproduction and adaptation in any medium and for any purpose provided that it is properly attributed. For attribution, the original author(s), title, publication source (PeerJ) and either DOI or URL of the article must be cited.
License URL: https://creativecommons.org/licenses/by/4.0/

Keywords: Affective, Cognitive, Medicolegal, Behavior, Medical students

Funding: The authors received no funding for this work.

==============================
Medical students attending university for the first time experience a new environment, full of significant social, cultural, and intellectual challenges. Moreover, drug abuse and bullying among university students are major global concerns. The aim of the current study was to assess the impact of medicolegal issues on undergraduate and postgraduate students. It is a cross-sectional survey-based study, with each set of questions investigating cognitive functions, aggression, personality, and exposure to medicolegal issues. Males and those with a chronic disease have been significantly exposed to medicolegal issues; exposed students were significantly older than nonexposed ones. The scores of aggression were significantly higher among exposed and male students. The cognitive scores were higher for the students from rural areas than in urban areas, and females were more neurotic than males. The current study recommends conducting campaigns to educate university students on the importance of formally disclosing unethical behaviors and listening to the victims to facilitate overcoming their negative feelings. As many victims feel comfortable disclosing victimization to their friends, we recommend conducting peer educational programs to help friends support their colleagues regarding unethical misconduct.

Introduction

Medical students attending university for the first time experience a new environment, full of significant social, cultural, and intellectual challenges. A favorable and safe learning environment is key to enhancing education and learning. The cognitive profile of the university student is affected by multiple medicolegal situations arising in the college community. “Medicolegal situations” refer to situations at the intersection of medicine and law (Ameis et al., 2013). In the context of university students, it could encompass a variety of scenarios. These might include issues related to health services provided on campus, medical ethics, student rights, or legal implications arising from medical incidents involving students, such as various traumatic events, substance or drug abuse, and professor-student bullying or intimidation (Frazier, 2009).

The experiences of bullying and harassment experienced by medical students have become important research subjects (Wood, 2006; Phillips et al., 2019). The most common traumatic events reported in university life and associated with higher distress levels are loss of close people, exposure to domestic violence, and sexual assaults (Frazier, 2009).

Moreover, drug addiction among university students has become a prominent issue in many countries. Exposure to common substances of abuse, including alcohol, tobacco, cannabis, and other psychoactive substances, is dramatically increasing. Substance or drug abuse adversely affects university students’ physical and mental health and academic life, causing absence, retarded academic achievement, and collision with others. Peer pressure, the desire to experiment, and the absence of proactive programs to combat drug abuse and raise awareness of its dangers can contribute to students’ tendency to abuse drugs or substances. The lack of clear legislation concerning age restrictions and awareness about substance abuse has short- and long-term impacts and can also exacerbate the problem (Tamrat, 2018). Bullying among students is another growing phenomenon in universities. Abusive behavior among young people is a severe problem exacerbated by the rapid advances in electronic communication. This problem reflects peer-on-peer violence, intimidation, social exclusion, and victimization. The effect of bullying on the mental well-being of the victim and his cognitive profile is exceptionally negative (Cowie & Myers, 2015). Studies have shown that high rates of college students being bullied by professors. They perceive professor/instructor bullying but this may not be sufficiently addressed (Marraccini, Weyandt & Rossi, 2015). Concerning harassment and bullying, it was considered that they do not occur in isolation, in this regard, Simpson & Cohen (2004) stated that “Both bullying and harassment are forms of unwelcome conduct that injures the person being targeted and is not appropriate in a professional or work environment.” Furthermore, they suggested that negative behavior directed towards an individual on the basis of their race, sex, disability, age, or sexual orientation is linked to harassment and provocation. By comparison, they understand bullying as operating on a more personal level and being contingent upon factors such as personality qualities, work position, or competency levels (Simpson & Cohen, 2004).

Despite the reported cases of the impact of exposure to medicolegal issues, there is a need to investigate the relationship between the neuropsychiatric makeup of the person and how he perceives this insult. Additionally, it is a challenging task to determine the trigger and the result of this interaction. It is important to combine different aspects of specialties and to use standard tools to have an accepted medicolegal conclusion shared by different stakeholders involved in healthcare (Morena et al., 2022).

The research question that we tried to find an answer to was: Are there any differences in cognitive and affective abilities between students who had been exposed to any medicolegal issue and other students without having this experience? Accordingly, the primary objective of this study was to assess the cognitive function of the students exposure to medicolegal problems, knowledge, and attitude to medicolegal perspective, and the secondary objective was to investigate the correlations between the collected data and the impact of having a history of being exposed to any illegal or unethical event on the higher functions of the brain that will, in turn, affect his academic achievement and career.

Subjects and methods

Participants, study setting and design

A cross-sectional survey-based study was conducted among undergraduate and postgraduate medical and paramedical students enrolled mainly from Jordan, Egypt, and the USA over one year (from the end of 2021 till the end of 2022). Eligible participants who met the inclusion criteria and completed the consent form were enrolled in the study. The study protocol was approved by the institutional review board of the Hashemite University (approval number: 5/12/2020/2021).

Sampling and sample size

Convenience sampling was adopted to approach the highest number of students. However, to ensure the accuracy of the obtained findings, the minimum sample size was calculated using The Raosoft statistical package Software. To achieve a 95% confidence level and a margin of error of 8%, a minimum sample size of 250 students was needed for a single proportion using the large sample normal approximation. To compensate for possible losses, 10% had been added, so the total targeted sample size was 275 cases. The outcomes of this study would be generalized to similar student communities and not the whole population.

Inclusion and exclusion criteria

Inclusion criteria included medical and paramedical undergraduate and postgraduate students actively enrolled in relevant educational programs during the study period. However, those who did not actively register (blocked or suspended students), others who refused to participate, and those who submitted incomplete responses were excluded.

The study aims and objectives were thoroughly explained to the participants, who were informed about their rights to withdraw without consequences. Responses were handled anonymously according to the Declaration of Helsinki and its later amendment in 1964.

Piloting

The adopted questionnaires consisted of sets of questions. Each set investigated aspects of the functions that aimed to be evaluated by the study. Questions were standardized in the form of dichotomous and rating scale questions. The preliminary version of the adopted survey was piloted on 30 students. The pilot study aimed to ensure all items were clear and easily understood. Results of the pilot study showed that all items were clear, and the estimated time for completing the survey was about 7–10 min. The stability of the piloted survey was assessed by calculating Cronbach’s alpha (0.87). However, these 30 responses were excluded from the analyzed responses.

The adopted survey consisted of five parts, as follows

The study surveys included Part 1, Cognitive aspects: Executive Skills Questionnaire (ESQ) (Strait et al., 2020); Part 2, Aggression measurement: The Buss–Perry Aggression Questionnaire (Gerevich, Bácskai & Czobor, 2007); Part 3: Personality assessment: A short questionnaire for measuring two dimensions of personality (Eysenck, 1958); Part 4: Exposure to the medicolegal problem: Finkelhor’s questionnaire for measuring sexual abuse (Aboul-Hagag & Hamed, 2012); and Part 5: Knowledge about the medicolegal problem.

The cognitive questionnaire included questions that assess working memory, planning and prioritizing, organization, metacognition, emotional control, and sustained attention. The aggression scales assessed both the physical and verbal responses of the person if he/she was provocated or threatened. The personality questionnaire was used to obtain the neuroticism and extraversion score of each participant. The other questionnaires examined the medicolegal issues the participants were exposed to e.g., sexual harassment, using illegal drugs, and bullying by colleagues, as well as the perception of students to exposure to these medicolegal issues.

The questionnaires used have already been validated by other studies on similar populations. Additionally, both content validity and construct validity were applied to the questions asked to the participants.

Questions posted outside of the standardized questionnaires included demographic data, level of education, specialty, and whether having chronic illnesses or not.

The adopted web-based anonymous survey was distributed using Google Forms. The authors used social media platforms like Facebook, WhatsApp, and Instagram to disseminate the survey link. Participants were informed about the purpose and advantages of the analysis following a thorough description of the study’s objectives and the anonymous data collection process. It was optional to take part in the survey. The respondents gave their permission for the survey to be conducted (Supplemental File). Participants were informed about the purpose and advantages of the analysis following a thorough description of the study’s objectives and the anonymous data collection process. It was optional to take part in the survey. The respondents gave their permission for the survey to be conducted (Supplemental File).

Scale scoring

Responses of the study participants to each cognitive, aggression, and personality measure were counted, and the sum was the participant’s score in each measure. For the cognitive questionnaire, the scoring was as follows: seldom = 0, rarely = 1, sometimes = 2, and frequently = 4. Then, the responses to the 30 questionnaire questions were summed up to get each student’s overall cognitive assessment score, with higher scores denoting the presence of more cognitive issues. Yes or no responses to the aggression and personality measures were coded as 1 and 0, respectively, and summed up.

Determining exposure

The study participant determined if he/she was exposed to one or more of the medicolegal issues, which included exposure to sexual harassment, pornographic pictures or movies, cheating in exams, stealing anything from the university or colleagues, any form of bullying by colleagues or professors, or blackmailing.

Statistical analysis

Categorical sample characteristics, e.g., sex and residence, were presented as a number and percentage, while continuous ones, e.g., age and cognitive scales, were presented as a mean and standard deviation. The chi-square test was used to examine the relationship between categorical variables. In contrast, non-parametric tests, e.g., Kruskal-Wallis and Wilcoxon signed-rank tests, were used to examine the group differences of non-normally distributed variables, e.g., age and cognitive scales. These continuous variabilities were assessed for normality, and all were non-normally distributed. Significant variables in the univariate analysis of exposure status and scores of cognitive, aggression, and personality measures were pooled in linear or logistic regression models accordingly. The regression coefficients, standard errors, test statistics, and p-value were reported for each regression model. The p-value was considered significant if less than 0.05. Odds ratios and confidence intervals were also reported for regression models.

Results

Study sample characteristics

As shown in Table 1, most study participants were females (75%), with a mean age of 20.8 years. Most of the group resided in urban areas (85%). The table also shows that about 90% were from Jordan, while the rest were from other countries, including Egypt and the United States of America. Also, about 97% of the group were students in the College of Medicine. A similar percentage were undergraduate students (94%), and most (92%) were in the first four years of their school. About 91% of the study sample lived with their family, while only about 9% lived away from families. Those who lived away from their families lived alone (52%) or with friends (48%). About 13% of the study sample had chronic diseases, including diabetes, heart disease, hypertension, and renal diseases.

Table 1 Study sample characteristics.

Factor	Frequency	Percent	Cumulative frequency	
Sex				
Female	209	74.64	209	
Male	71	25.36	280	
Residence				
Rural	43	15.36	43	
Urban	237	84.64	280	
Country				
Egypt	19	6.79	19	
Jordan	251	89.64	270	
USA	4	1.43	274	
Others	6	2.14	280	
College				
Medicine	271	96.79	271	
Pharmacy	3	1.07	274	
Physiotherapy	1	0.36	275	
Others	5	1.79	280	
Level of education				
Postgraduate	16	5.71	16	
Undergraduate	264	94.29	280	
Living with family				
No	25	8.93	25	
Yes	255	91.07	280	
Having a chronic disease				
No	245	87.50	245	
Yes	35	12.50	280	
Age	Mean = 20.8	SD = 3.3	274	

Exposure to medicolegal issues by sociodemographic characteristics

Figure 1 shows that males (73.2%) and those who reported having a chronic disease (74.3%) were significantly exposed to medicolegal issues more than females (51.7%) and those who did not report having a chronic disease (54.7%) (p < 0.05), respectively. In addition, the Kruskal-Wallis test showed that exposed students (mean age = 21.1 years) were significantly older than nonexposed ones (mean age = 20.5 years, p < 0.001). On the other hand, there were no significant differences between exposed and nonexposed students by place of residence, country of origin, level of education, or the status of living with the family (p > 0.05).

Figure 1 Exposure to medicolegal issues by study characteristics.

As shown in Table 2, the logistic regression analysis model, including students’ sex, age, and history of having a chronic disease, was overall significant in predicting the status of student exposure to medicolegal issues (Chi-Square = 16.4, p = 0.001). After controlling for other variables, the sex of students was the only significant factor in predicting the student status of exposure (Wald Chi-square = 9.4, p = 0.002). The odds of being exposed to medicolegal issues among males are 2.5 times the odds of being exposed among females (CI [1.4–4.6]).

Table 2 Logistic regression analysis of factors predisposing to exposure to medicolegal issues.

Variables	b (SE)	Wald Chi-square	p-value	Odds	CI	
Intercept	−1.2 (0.9)	1.8	0.2			
Sex (male vs females)	0.9 (0.3)	9.4	0.002	2.5	[1.4–4.6]	
Age	0.06 (0.04)	1.8	0.2	1.1	[0.97–1.2]	
History of chronic disease	0.7 (0.4)	2.8	0.09	2.0	[0.9–4.6]	
Model Chi-square = 16.4, p = 0.001	

Perception of medicolegal issues by exposure status

Figure 2 demonstrates that a more significant percentage of exposed students (34.4%) were subjected to pressure or stress preventing them from disclosing criminal suspicion than the percentage of nonexposed students (14.2%, p < 0.001). There were no significant differences between exposed and nonexposed students in the percentages of those who think they must notify police authority immediately or notify relatives prior to police notification about medicolegal problems (p > 0.05).

Figure 2 Perception of medicolegal issues by exposure status.

Cognitive, aggression, and personality measures by exposure and characteristics of study participants

Table 3 shows that the score of the aggression scale was significantly higher among exposed (4.3) and male (4.8) students than the score among nonexposed (3.2) and female (3.4) students (p < 0.001), respectively. Regarding the cognitive scale, it was found that students from rural areas (45.9) scored higher than those from urban areas (42.3, p = 0.009). For personality measures, it is only that females (4.3) were more neurotic than males (3.8, p = 0.02).

Table 3 Score of cognitive, aggression, and personality measures by exposure and characteristics of study participants.

Factor	Cognitive
Mean (SD)	Aggression
Mean (SD)	Neuroticism
Mean (SD)	Extraversion
Mean (SD)	
Exposure					
No (n = 120)	42.0 (9.9)	3.2 (2.1)	4.3 (1.8)	3.6 (1.6)	
Yes (n = 160)	43.5 (9.1)	4.2 (2.1)	4.1 (1.6)	4.0 (1.5)	
Sig (Mann-Whitney test)	0.2	<0.001	0.3	0.1	
Sex					
Female (n = 209)	42.3 (9.9)	3.4 (2.1)	4.3 (1.7)	3.8 (1.6)	
Male (n = 71)	44.5 (8.0)	4.8 (2.0)	3.8 (1.7)	3.9 (1.5)	
Sig (Mann-Whitney test)	0.08	<0.0001	0.02	0.4	
Residence					
Rural (n = 43)	45.9 (10.0)	4.0 (2.4)	4.4 (1.8)	3.9 (1.5)	
Urban (n = 237)	42.3 (9.3)	3.7 (2.1)	4.2 (1.7)	3.8 (1.6)	
Sig (Mann-Whitney test)	0.009	0.4	0.3	0.5	
Country					
Jordan (n = 251)	43.3 (8.7)	3.8 (2.2)	4.2 (1.7)	3.8 (1.5)	
Others (n = 29)	38.7 (13.4)	3.8 (2.1)	4.3 (1.4)	3.5 (1.6)	
Sig (Mann-Whitney test)	0.1	0.8	0.8	0.3	
Level of education					
Postgraduate (n = 16)	37.5 (15.2)	4.6 (2.9)	4.2 (1.9)	3.6 (1.9)	
Undergraduate (n = 264)	43.2 (8.9	3.7 (2.1)	4.2 (1.7)	3.8 (1.5)	
Sig (Mann-Whitney test)	0.2	0.2	0.9	0.5	
Living with Family					
No (n = 25)	41.0 (12.1)	3.8 (2.4)	4.2 (1.8)	3.9 (1.6)	
Yes (n = 255)	43.0 (9.2)	3.8 (2.1)	4.2 (1.7)	3.8 (1.6)	
Sig (Mann-Whitney test)	0.6	0.9	0.9	0.8	
Having a chronic disease					
No (n = 245)	43.1 (9.2)	3.7 (2.1)	4.2 (1.7)	3.8 (1.5)	
Yes (n = 35)	41.3 (11.4)	4.6 (2.5)	4.1 (1.8)	3.9 (1.8)	
Sig (Mann-Whitney test)	0.7	0.06	0.7	0.6	
Age (n = 274) (Spearman Correlation)	−0.01, 0.8	0.8	0	−0.1	
Sig	0.8	0.8	0.9	0.3	
Note:

Bold p-values are significant.

The linear regression analysis model, which included students’ exposure status, sex, and history of having a chronic disease, was overall significant in predicting the score of the aggression scale of the study sample (F value = 12.2, p < 0.001). Being exposed to medicolegal issues (t value = 3.0, p = 0.003) and a male (t value = 4.0, p < 0.001) were significant predictors of the aggression scale score. About 12% of the changes in the score of the aggression scale were related to the changes in the variables in the regression model (R-Square = 0.12) (Table 4).

Table 4 Linear regression of aggression scale.

Variables	b (SE)	b (95% CI)	t value	p-value	
Intercept	3.0 (0.2)	3.0 [2.2–3.8]	15.4	<0.001	
Exposure (Yes vs No)	0.8 (0.2)	0.8 [0.3–1.3]	3.0	0.003	
Sex (male vs females)	1.1 (0.3)	1.1 [0.6–1.7]	4.0	<0.001	
History of chronic disease	0.6 (0.4)	0.6 [−0.1 to 1.4]	1.7	0.08	
Model F value = 12.2, p < 0.001	
R-Square = 0.12	
Note:

Bold p-values are significant.

Discussion

Domestic violence is a frank violation of human rights. It causes dramatic health consequences recognized worldwide as a public health burden. No country is excluded from the risks and maleffects of violence. Furthermore, one of the global causes of death and disability is violence. It covers coercive and controlling behavior as well as financial, emotional, psychological, sexual, and physical abuse (Guy, Feinstein & Griffiths, 2014). Exposure to medicolegal concerns, especially among youth (undergraduate students), significantly affects their personality development and the configuration of their professional profile. Therefore, the current study investigated the exposure to illegal insults, considering the individuals’ personality, behavioral, and cognitive processing in medical field students (medical and paramedical, undergraduate and postgraduate students).

According to the literature, there is a major difference in the definition of domestic abuse among different cultures (Fernandez, 2006). In our study, most participants were females (75%) with a mean age of 20.8 years, and the females exposed to medicolegal issues, including different types of harassment, represent 51.7%. In addition, the majority resided in urban areas (85%), and about 90% were from Jordan, while the rest were from other countries. The oriental nature of Jordanian social culture embraces violence against children or women as a form of discipline, and this attitude is condoned and encouraged by cultural and social standards. Furthermore, earlier research (Gharaibeh & Al-Ma’aitah, 2002; Okour & Hijazi, 2009; Al-Badayneh, 2012) discovered that exposure to family violence, whether real or witnessed, has a major impact on university students.

In agreement with our study, a cross-sectional study was conducted among voluntary healthcare students 18 years and over in the Health Campus at Rouen and nursing schools in Normandy, France. There were five gender-based violence (GBV) kinds documented. With a mean age of 20.8 years, females comprised most of the studied sample. More than 40% of students have experienced at least one form of GBV since the commencement of their healthcare study. Regardless of the kind of GBV, the perpetrators were other students, healthcare professionals, and patients (Tavolacci et al., 2023).

According to Faith et al. (2015), kids and teenagers with long-term health conditions are more likely to experience bullying because peers may perceive them as being different due to symptoms of the illness or treatment plans (Talmon, 2010). Pinquart (2013a) likewise showed that adolescents suffering from chronic illnesses are more likely to experience diminished social and academic functioning, perhaps leading to negative reactions from their peers. Furthermore, these kids may be more vulnerable to bullying because of their psychological vulnerabilities, which include stigmatization and low self-esteem (Pinquart, 2013b).

Sexual violence is a medicolegal issue that medical students may encounter. Sexual violence is defined by the Centers for Disease Control and Prevention (CDC) as “an attempt or commission of a sexual act by someone without the victim’s free permission or against an individual who is incapable of consenting or refusing” (Jatmiko, Syukron & Mekarsari, 2020). Sexual harassment is included within sexual violence and was reported as a critical public health concern among university students (Bonar et al., 2022).

The current study conveyed that being male may significantly predict exposure to medicolegal issues, including sexual harassment. In partial agreement with the current study, it was reported that among university students, males are more likely to commit sexual perpetration (Salazar et al., 2018). Another study reported that the risk of sexual perpetration among males is directly related to alcohol and drug misuse (Naguib et al., 2008). The discrepancy between the current study and previous studies could be explained by the fact that the current study dealt with being a victim or perpetrator equally when asking questions on exposure to medicolegal issues. Moreover, there is a scarcity of studies on male sexual victimization and female sexual perpetration (Xiao & Wong, 2013).

Consistent with the current study, the high prevalence of substance abuse among male university students was previously reported. Naguib et al. (2008) conducted their study among Egyptian university students and reported that 93.1% of cannabinoid abusers were male compared to 9.6% of females. Low socioeconomic standards and living in rural areas were significant predictors of cannabinoid abuse.

As per the current study, it was agreed that male university students more frequently committed bullying. However, male and female students were equally victimized by bullying (Cowie & Myers, 2015). Nevertheless, due to the advancement of technology, bullying forms varied. Facebook and cyberbullying were reported to be higher among male university students than females (Xiao & Wong, 2013).

Literature reported conflicting findings regarding gender-based variations in cheating among university students. Brown & Choong (2003) reported equal cheating practices among male and female university students. Similarly, a previous study among Jordanian medical students regarding cheating in online exams during the last pandemic revealed no significant gender variations (Elsalem et al., 2021). Nevertheless, others mentioned that males were significantly more cheaters than females (Yazici et al., 2023), whereas Graham et al. (1994) reported higher cheating prevalence among female university students. However, in agreement with the current study, a meta-analysis of 48 studies reported that females were less exposed to unethical behaviors, including cheating. They tended to have positive attitudes toward cheating (Whitley, Nelson & Jones, 1999).

Indeed, exposure to sexual harassment, pornography, bullying, illicit drug use, and other unethical behaviors among university students are inseparable. An Italian study conducted among 12th-grade students and students aged 18–25 conveyed that all males and 67% of females had watched pornography (Romito & Beltramini, 2011). High school females involved in sexual practice were previous victims of bullying and were mainly alcohol addicts. Similarly, males were previous bullying perpetrators and marijuana users (Woodward, Evans & Brooks, 2017).

Furthermore, Salazar et al. (2018) found that before starting college, young men who said they watched more sexual media, drank a lot, had hypermasculine beliefs, and had peers who supported sexual violence were more likely to have a history of committing sexual violence while in college. A previous study reported that alcohol and illicit drug usage are significantly associated with sexual violence (perpetration or victimization) (Krebs et al., 2009).

The current study revealed that more university students exposed to medicolegal issues were subjected to pressure/stress, preventing them from disclosing criminal suspicion. In agreement with the current study, formal university sexual assault disclosure resources are significantly underutilized (Holland & Cortina, 2017). Multiple social factors shape the ability of students to disclose their exposure to medicolegal issues, particularly sexual violence, which fuels the perpetration against them (Jatmiko, Syukron & Mekarsari, 2020). Stigma and shame were significant reported barriers to disclosing exposure to sexual assault. Other reported factors were guilt, self-blame, and fear of breaking confidentiality by informing friends and family. Moreover, few victims felt that victimization was not severe enough to be reported (Stoner & Cramer, 2019).

The current study’s findings regarding aggression aligned with the vast body of literature highlighting the great liability of developing aggressive, violent attitudes in youth who had grown up in a domestic violence atmosphere (Davis & Lindsay, 2004).

Complex trauma describes children exposed to multiple painful, traumatic events, often with invasive, interpersonal nature, mostly affect youth negatively and may give rise to criminal behavior. Delinquent youth may suffer particularly adverse effects from witnessing or being exposed to violence. The risk of juvenile crime was more significantly correlated with exposure to violence rather than with a history of abuse, the severity of traumatic stress symptoms, and the likelihood of suicidal behavior and substance addiction among incarcerated juvenile offenders (Naguib et al., 2021; Hoff et al., 2009).

Additionally, exposure to widespread incidents that constitute complex trauma is associated with changed cognitive information processing, expectations, and configuration, which may cause young individuals to become aggressive (Krug et al., 2002), reactive aggressiveness is more likely in people who allow themselves to be victimized or who harbor strong feelings of guilt and self-blame. Furthermore, complicated trauma might result in peer interaction with delinquents or their encouragement. Because behavioral disorders tend to cluster together, being around delinquent peers may enhance the predisposition toward psychosocial impairment and violence (Yuan et al., 2014).

Moreover, the present study showed that male students were more aggressive than female students, which agrees with previous studies that explain male physical aggressiveness higher than female (Eagly, Wood & Diekman, 2000). For this disparity, there are two major and competing interpretations. The first hypothesis, social role theory, contends that sex variations in physical aggression result from indoctrination into gender roles that prescribe the use of aggression and violence differently for males and females. In essence, social role theorists contend that men are more inclined to engage in physical aggression as a result of cultural focus on traditional dominant and competitive roles (Alsawalqa, Alrawashdeh & Hasan, 2021).

Women, on the other hand, are socialized into more docile and mild roles that discourage aggression. Later versions of the theory incorporated physical sex differences as role restrictions. Eagly, Wood & Diekman (2000) suggest that developed physical disparities between males and females are responsible for the emergence of stereotyped divisions of labor and social roles that influence the proclivity for physical aggressiveness.

The theory described in the previous paragraphs may be referred to as Man box (Socially imposed expectations, views, and behaviors that are deemed “manly” and/or “real man’s” behavior, such as superiority, cruelty, emotional suppression, lack of physical intimacy with other males, and socially aggressive and/or dominant behavior.)and it was already studied by Alsawalqa, Alrawashdeh & Hasan (2021) in Jordanian society where most of our participants belong. The study confirmed that in Jordan the Man Box ideas were agreed by most of the males and females who participated in their study.

The second theory stems from evolutionary models of sexual selection, which state that because males invest less in reproduction than females do, they face more competition for successful reproduction. Because of this, men have developed a variety of physical traits and mental strategies that make it easier for them to engage in combat over resources and mates (Archer, 2009). Adolescent males utilize physical aggressiveness as a means of establishing social authority and engaging in productive competition for resources and status, including popularity, reputation, and the ability to enter into partnerships (Hoff et al., 2009).

The study findings revealed that students from rural areas had higher cognitive scores than students from urban areas. These are not in line with Melrose et al. (2013), who discovered that lower childhood socioeconomic status was linked to higher rates of cognitive decline and worse late-life cognitive performance as assessed by executive functioning, episodic memory, and semantic memory. Greenfield & Moorman (2018)’s analysis of data from 5,074 Wisconsin Longitudinal Study participants revealed a substantial correlation between early SES and cognitive function.

Moreover, our study findings showed that female students had been found to have more neuroticism than male students, and this is consistent with the literature findings; for instance, according to Costa, Terracciano & McCrae (2001), girls exhibit higher degrees of neuroticism than boys do.

Adolescence is characterized by androgen hormones and physical growth. Different hormone functions affect mood differently in men and women. Hormonal functions have an impact on psychopathology genetic predisposition and personality as well (Pfeifer & Allen, 2021). Adolescent interest, activity, and propensity toward violence are also explained by androgen hormones (Ramirez, 2003). Compared to boys, girls exhibit more frequent hormonal shifts (Djudiyah, Harding & Sumantri, 2016). Girls typically experience monthly menstruation which correlates with neuroticism and impaired emotional control as reported by Wu, Zhou & Huang (2014).

Based on the aforementioned speculations, it is a fact that violence is a multifaceted social and public health issue that needs comprehensive intervention strategies. Semahegn et al. (2019) reported some tools considered effective in combating violence and the other phenomena discussed in the current study. Balanced powers in relationships, reducing alcohol consumption, ensuring good mental health, and advocating for violence screening are crucial initial steps. More efforts are needed to dispel myths, misconceptions, traditional norms, and beliefs of the community. Community orientations to avoid traditional gender norms, ensure gender equity, and increase awareness about the consequences of violence are mandatory.

Strength and limitations

The strength of the current study resides in its multinational, multi-institutional nature. Convenience sampling may introduce selection bias; however, our sample reflected the student demographics in Jordan, as appears in Table 1. The table shows that most students in the medical are paramedical colleges were females, “living in urban communities, and living with their families”.

The limitations of the study include the bias of the self-reporting survey. In addition, the R square of the regression model is only 0.12, which indicates the limited explanation of the changes in the aggression score because of the changes in the predicting variables included in the regression model.

Conclusions

Males and those with a chronic disease were significantly exposed to medicolegal issues more than females and those who did not have a chronic disease, respectively. Exposed students were significantly older than nonexposed ones, with the sex of the students as the only significant factor in predicting the exposure status.

Exposed students were more stressed to disclose criminal suspicion than nonexposed students. Besides, the aggression scale score was significantly higher among exposed and male students than among nonexposed and female students, respectively. The score on the aggression scale was significantly higher among exposed and male students than the score among nonexposed and female students, respectively.

The cognitive scale was higher in the students from rural areas than in urban areas. On the other hand, looking into the personality contribution, females were more neurotic than males.

The current study shows that conducting campaigns educating university students on the importance of formal disclosure of unethical behaviors, including sexual harassment, is warranted. Listening to the victims and helping them to overcome the negative feelings and stigma could be the first step in the healing process. To develop the formal disclosure process, one should work systematically to acknowledge the courage of telling the event, respecting the pain, assuring confidentiality, and advising of authority reporting in required cases.

As many victims feel comfortable disclosing victimization to their friends, conducting peer education programs may help friends support their colleagues regarding sexual harassment, alcohol abuse, and other unethical misconduct.

There is a need to amend and enforce the existing laws and formulate new policies to mitigate and respond to violence. Also, we advise involving victims of medicolegal issues with an educational approach to cope with posttraumatic psychological consequences and minimize future violence.

Nevertheless, we encourage future research to reveal unknown determinants of exposure to medicolegal issues among university students. In addition, further studies are recommended with the inclusion of systematic surveys that enable greater participation of the male population and ensure more homogeneous cases.

Supplemental Information

Supplemental Information 1 Questionnaire (English).

Click here for additional data file.

Supplemental Information 2 STROBE checklist.

Click here for additional data file.

Supplemental Information 3 Participant responses.

Click here for additional data file.

Additional Information and Declarations

Competing Interests

Author Contributions

Human Ethics

Data Availability

The authors declare that they have no competing interests.

Shaimaa Nasr Amin conceived and designed the experiments, performed the experiments, authored or reviewed drafts of the article, and approved the final draft.

Ghada Nazar Al-Jussani performed the experiments, authored or reviewed drafts of the article, and approved the final draft.

Sherif S. Hassan conceived and designed the experiments, authored or reviewed drafts of the article, and approved the final draft.

Asmaa F. Sharif performed the experiments, authored or reviewed drafts of the article, and approved the final draft.

Ahmed A. Ismail analyzed the data, prepared figures and/or tables, authored or reviewed drafts of the article, and approved the final draft.

Dina Badie Taher performed the experiments, authored or reviewed drafts of the article, and approved the final draft.

Iman Aolymat performed the experiments, authored or reviewed drafts of the article, and approved the final draft.

Gehan El-Akabawy analyzed the data, authored or reviewed drafts of the article, and approved the final draft.

Abeer Ahmed Zayed conceived and designed the experiments, authored or reviewed drafts of the article, and approved the final draft.

The following information was supplied relating to ethical approvals (i.e., approving body and any reference numbers):

Institutional Review Board of the Hashemite University.

The following information was supplied regarding data availability:

The raw data is available in the Supplemental Files.

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
