# Peer review of "Evaluating cognitive and affective abilities among medical students: behavioural and medicolegal perspectives"

_PeerJ, doi:10.7717/peerj.16864_

## Round 0.1 · original submission · Major Revisions

I have now received the reviewers' comments on your manuscript. They have suggested some revisions to your manuscript. Therefore, I invite you to respond to the reviewers' comments and revise your manuscript.

Reviewer 1 ·

Basic reporting

The introduction provides relevant background on the issues explored in the study, such as trauma, substance abuse, and bullying among university students. However, it would benefit from more focus on the specific research questions and hypotheses being examined in this study. The introduction should clearly state the rationale and objectives for this particular study.

The discussion section interprets the main findings and compares them to prior literature. However, the strengths and limitations of the study should be discussed more thoroughly before conclusions. The conclusions would also benefit from more specific implications and recommendations based on the study results.

Experimental design

The cross-sectional survey design is appropriate for the study objectives of assessing exposures to medicolegal issues and evaluating cognitive/affective factors among students. However, there are some limitations with the experimental design:

- Convenience sampling was used to recruit participants, which can introduce selection bias. The limitations of this sampling method should be addressed

- The study has a relatively small sample size of 280 students from a few universities. The authors should discuss how this may limit generalizability of the findings

Validity of the findings

There is limited information provided on the survey development and piloting process. More details are needed on assessing reliability and validity of the survey tools used.

The relatively small convenience sample from only a few academic institutions may not be representative of the broader student population. Cautions regarding generalizability of the results should be added.

Overall, the limitations discussed above preclude strong conclusions about causal relationships between variables. The conclusions drawn should better reflect the exploratory nature of this study.

Additional comments

Tip that could really streghten the paper...

Findings are mainly based on bivariate comparisons between exposure and outcome variables. Multivariable analyses adjusting for potential confounders would strengthen the validity of results.
While bivariate analyses are informative, they do not control for potential confounding factors. For instance, the difference in aggression scores between exposed and non-exposed students could possibly be influenced by variables like gender, age, or other demographics. For example, the authors could conduct multiple linear regression for the outcome of aggression scores with predictors including medicolegal issue exposure, gender, age, and any other plausible confounders. The adjusted regression coefficient for exposure would estimate the effect of medicolegal issues on aggression independent of the other variables in the model. By controlling for likely confounders, the application of multivariable analyses helps isolate the relationship between the exposure and outcome and reduces the likelihood of spurious findings due to confounding. This would lend more validity to the study results and allow stronger conclusions about the true effects of medicolegal issue exposure on cognitive/affective measures. The lack of adjustment for confounders is a limitation of the predominantly bivariate approach used. Using more advanced multivariate regression models that include relevant confounders could strengthen the validity of the results. Multivariable analyses account for multiple factors simultaneously to estimate the independent association between the exposure and outcome after adjusting for potential confounders.

Reviewer 2 ·

Basic reporting

The article is written decently and quite clearly in defining the methods used for the investigation. Unfortunately, there is a lack of clarity regarding what the authors mean by "exposure to medicolegal issues”. Are they referring to victims or perpetrators? This is crucial, as the results depend on this information.

The introductory part should be improved, as there are few references to other studies in the literature, and, more importantly, the reasons for conducting the study are not well articulated.

Here are some critical points I would like to highlight:
52 medicolegal situations
What does "medicolegal situations" mean? Is it referring to criminal offenses, civil cases, administrative matters, etc.? It should be specified..


194 -196 “Only about 13% of the study sample had chronic diseases, including diabetes, heart disease, hypertension, and renal diseases”
Considering that it involves very young individuals, this seems to be a significant piece of information, so I would omit the term “only”.
203 “nonexposed ones (mean age = 20.5 years, p = 0.0001)” : Better, p<0.001
206 “living with the family (p > 0.0)” : Fix it

253-263 A mention should be made of gender-based violence, a topic of increasing interest that aligns with domestic violence but is not limited to it. It is indeed a subject that resonates strongly even among the younger population, as evidenced by various news cases, some of which have involved medical students [e.g., Morena, D., Di Fazio, N., La Russa, R., Delogu, G., Frati, P., Fineschi, V., & Ferracuti, S. (2022). When COVID-19 Is Not All: Femicide Conducted by a Murderer with a Narcissistic Personality “Masked” by a Brief Psychotic Disorder, with a Mini-Review. International journal of environmental research and public health, 19(22), 14826.]

264-267 I suggest making this sentence simpler and more straightforward.
296-298 I don't understand what the reference to pornography has to do with this.
319-321 Moreover, male university students using pornography were later diagnosed with sexual
aggression [22].
This is not true, namely, the cited article reports: “Before starting college, young men who reported more sexual media consumption, heavy episodic drinking, hypermasculine beliefs, and peers who endorsed SV were more likely to have a history of SV perpetration at college matriculation”. This is very different from what is assumed in the paper.

"Considering the geographical area of the study, it is useful to refer to the characteristics of the so-called 'Man Box' [Alsawalqa, R. O., Alrawashdeh, M. N., & Hasan, S. (2021). Understanding the Man Box: the link between gender socialization and domestic violence in Jordan. Heliyon, 7(10).]

390 ”literature findings;”
It lacks the bibliographic reference.

395 predisposition and personality as well.
It lacks the bibliographic reference.

396 Adolescent interest, activity, and propensity toward violence are also explained by androgen hormones.
It lacks the bibliographic reference.

397 – 399 Girls typically experience monthly menstruation and are more emotionally unstable. [46]
From which study do the authors correlate menstruation with neuroticism?

Finally, it would also be useful to reference the tools considered effective in combating IPV and the other discussed phenomena [e.g., Semahegn, A., Torpey, K., Manu, A., Assefa, N., Tesfaye, G., & Ankomah, A. (2019). Are interventions focused on gender-norms effective in preventing domestic violence against women in low and lower-middle income countries? A systematic review and meta-analysis. Reproductive health, 16, 1-31.]

Experimental design

Examples related to the questions posed outside of standardized questionnaires should be provided

136-142 Could you provide a brief description of each scale?

239 model (R-Square = 0.12) (table 4): Could you report the confidence intervals for B for Table 4?

427 limitations
The study has several limitations, particularly at the statistical level; the Model Chi-Square and R-square values are quite low, providing limited explanation for the differences

Validity of the findings

It should be clarified how this study has highlighted critical points that need further analysis, particularly through systematic surveys that enables greater participation of the male population and ensures more homogeneous cases

---

## Round 0.2 · accepted · Accept

In my opinion, this manuscript has been revised with attention to the reviewers' comments and can now be published.

Reviewer 1 ·

Basic reporting

The authors have satisfactorily addressed my comments

Experimental design

The authors have satisfactorily addressed my comments

Validity of the findings

The authors have satisfactorily addressed my comments